# Healthy Promotion for Fighting Metabolic Syndrome: Insights from Multi-Center HeRO-FiT Cohort

**DOI:** 10.3390/ijerph17155424

**Published:** 2020-07-28

**Authors:** Vincenzo Gianturco, Luigi Gianturco, Rebecca Regnoli, Bruno Dino Bodini, Maurizio Turiel, Martino Trapani, Francesco Bini, Giuseppe De Angelis

**Affiliations:** 1Hospital for the Elderly “Madonna del Divino Amore”, Via Casilina, 1839, 00132 Borghesiana, Rome, Italy; vincenzogianturco@gmail.com; 2ASST Rhodense, Cardiac-Rehab Unit, Passirana-Rho Hospital, Via Luigi Settembrini, 1, 20017 Rho Milan, Italy; giuseppe.deangelis@unimi.it; 3IRCCS Galeazzi Orthopedic Institute, Dietician Service, Via Riccardo Galeazzi, 4, 20161 Milan, Italy; rebecca.regnoli@gmail.com; 4ASST Rhodense, Pulmonology Rehab Unit, Passirana-Rho Hospital, Via Luigi Settembrini, 1, 20017 Rho Milan, Italy; bodinibruno@hotmail.com; 5IRCCS Galeazzi Orthopedic Institute, Cardiology Unit, Via Riccardo Galeazzi, 4, 20161 Milan, Italy; maurizio.turiel@unimi.it; 6ASST Rhodense, Public Health Division, Garbagnate Hospital, Via Carlo Forlanini, 20024 Garbagnate Milanese Milan, Italy; mtrapani@asst-rhodense.it; 7ASST Rhodense, Chief of Pulmonology, Garbagnate Hospital, Via Carlo Forlanini, 20024 Garbagnate Milanese Milan, Italy; fbini@asst-rhodense.it; 8ASST Rhodense, Chief of Cardiology Department Rho Hospital, Corso Europa, 250, 20017 Rho Milan, Italy; 9UniSR, Milan, Via Olgettina, 58, 20132 Milan, Italy

**Keywords:** metabolic syndrome, cardiovascular risk, health public, diet, physical activity

## Abstract

We know that metabolic syndrome (MS) is a modern cardiovascular (CV) “epidemic”, especially in western populations. MS is indeed strictly related to the risk of developing CV diseases (CVD) and/or diabetes. Therefore, the aim of our multi-center study was to promote a “healthy style” for fighting MS. Each participating center analyzed its own database of outpatients and globally we have pulled out 100 volunteers to participate in the study. Before starting, we collected their written consent. Enrolled subjects have not any history of overt CVD and/or diabetes, but they matched National Cholesterol Education Program/Adult Treatment Panel (NCEP/ATP) criteria for MS. After enrolment (t0), subjects were randomly divided into two homogeneous groups: a) only diet suggestions; b) both diet and exercise prescription. Later, we measured for each subject: blood pressure (BP), heart rate (HR), height, weight, body mass index (BMI), waist circumference (WC), waist hip ratio (WHR), six-minute walking test (WT6M), distance and common blood tests such as fasting plasma glucose, high-density lipoproteins (HDL) and triglycerides (T1 assessments). At six months (T2), the same parameters were measured and then statistical comparisons were performed. Attention to diet caused significant changes only in WC and WHR, whilst a coupling of exercise and diet revealed a statistically significant improvement in HR, BP, BMI, blood samplings and WT6M too. In conclusion, a healthy lifestyle should be more encouraged by physicians and/or collaborators (such as dieticians) operating in preventive settings. Diet and physical activity may be early useful strategies in the “battle” against MS even before any medication choices. Further studies will be necessary in order to better address the topic.

## 1. Introduction

The main objective of leading international societies of cardiology is the reduction of the cardiovascular (CV) burden and then CV disease (CVD) [1].

Nowadays, CVD remains the biggest cause of worldwide mortality despite improvements in outcomes especially in high-income countries [2]. In Europe, it still accounts for four million deaths per year (47% of all mortalities) with an estimated cost of €196 billion annually due to CVD [3].

Therefore, preventive measures are pivotal to contrast CV morbidity and mortality and in all the latest guidelines delivered by European Society of Cardiology (ESC), the attention to prevention has been growing [4,5,6]. In the last few decades, smoking legislation has decreased coronary artery disease (CAD) rates but other CV risk (CVR) factors (CVRF) such as obesity and diabetes mellitus (DM) have been substantially increasing [7].

Summing up, nowadays the main targets of primary prevention are blood pressure (BP), lipid and glucose levels and lifestyle. A cluster of CVRFs named metabolic syndrome (MS) encloses all the above-cited elements.

Thus, MS has become one of the most important enemies of Western countries and the efforts of healthcare providers should be gradually more addressed to combat it. MS is in fact a constellation of pathophysiological factors that increase the risk of developing DM type 2 and CVD [8]. Therefore, if we want to effectively contrast DM and CVD, the main early goal might be MS.

As described by Said et al., MS not only increases CV morbidity and mortality, but also affects the economic resources of all modern healthcare systems [9]. Thus, our efforts against MS will be increasingly important.

In literature we can find several definitions of MS and in the last few years attempts have been made to unify criteria in order to gain a consensus on its diagnosis [10]. The World Health Organization (WHO), International Diabetes Federation (IDF), National Cholesterol Education Program Adult Treatment Panel III (NCEP-ATP III) and the American Association of Clinical Endocrinologists (AACE) have proposed their diagnostic criteria or components of MS. In 2009, representatives of the IDF and the American Heart Association/National Heart, Lung, and Blood Institute (AHA/NHLBI)-ATP III Guidelines discussed resolving differences between definitions of MS, unifying the criteria [11]. The definition of MS according to the unification of criteria (harmonizing the metabolic syndrome) is used for many international works and publications and will be extensively described in methods.

The concept of MS was carried out in a progressive manner. Its clinical importance is enormous due to the fact that it can early identify patients (pts) with a high risk of suffering some CVD and/or DM, allowing a preventive intervention [12,13,14]. Identifying early stages of organ damages may reduce global CV and/or metabolic risks such as preventing worst complications and fatal events.

In several studies (both observational and/or interventional), physical activity improved fitness mitigating MS; even if it is still controversial, there is the possibility of normalizing insulin resistance, lipid disorders and obesity [15]. Therefore, the relationship between MS and CVR is far from defined.

By starting with this evidence, we have decided to design a preventive strategy to fight MS choosing a cohort of outpatients. The details of our study, which is named HeRO-FiT (health promotion for fighting metabolic syndrome), will be explained in the following section.

## 2. Methods

### 2.1. MS Definition

Summing up, the referral and/or standardized definition of MS requires the presence of three of the following five criteria [16]:Elevation of fasting blood glucose (≥100 mg/dL) or receiving antidiabetic treatment with insulin or oral antidiabeticsElevation of systolic blood pressure (SBP) ≥ 130 mmHg or diastolic blood pressure (DBP) ≥ 85 mmHg or receiving antihypertensive drug treatmentHigh-density lipoprotein (HDL) cholesterol values <40 mg/ dL (men) or <50 mg/dL (women)Triglycerides ≥ 150 mg/dL or under treatment with specific lipid lowering agentsAbdominal perimeter ≥ 102 cm (men) or ≥88 cm (women)

### 2.2. Experimental Design

Initially, we have retrospectively analyzed the outpatient databases of four different settings operating in CV Prevention and Risk Management included in the National Health Service (NHS) and extracted the first 100 consecutive schedules (about 25 for each center) of subjects (aged > 18 y/o) responding to MS criteria just shown. After this preliminary choice, we enrolled (t0) 100 volunteers who gave their written consent to the protocol. Next, enrolled subjects were randomly divided into two groups:(a)50 outpatients were directed and monitored only on a healthy diet (according to ESC guidelines about CVD Prevention) [17] and identified them as group “D”;(b)50 outpatients were directed and monitored both on a healthy diet and on an aerobic continuative exercise program (details in the following, specific paragraph) and identified them as group “DE”.

Two groups were evaluated thirty days post-enrolment (T1) for body max index (BMI), waist circumference (WC), waist hip ratio (WHR), BP, heart rate at rest (HR), 6-min walking test (WT6M) and common blood samples.

Two subjects from group D and one from group DE were lost at the first follow-up (FUP), identified as T1, for withdrawal of informed consent (Figure 1).

After 6 months (T2), the same measurements were repeated.

All authors periodically took turns in interviewing by phone-calls participating to the study in order to assess the right adherence to protocols (D and/or DE) and/or for motivating them.

The protocol was performed in accordance with the Declaration of Helsinki [18] and met the ethical standards for Sport and Exercise Science Research.

Finally, this study also did not provide any pharmacological prescriptions and/or invasive procedures.

### 2.3. Population Exclusion Criteria

Enrolment excluded pts with a known history of CAD, proven hypertension (HT), overt diabetes, heart failure (HF), chronic kidney disease (CKD), active cancer and any previous hospitalization and/or event due to CVD.

### 2.4. Exercise Prescription (only Group “DE”)

We recommended at least 150 min a week of aerobic moderate exercise such as walking briskly (4.8–6.5 km/h), slow cycling (15 km/h), painting/decorating, vacuuming, gardening (mowing the lawn), golf (pulling clubs in a trolley), tennis (doubles), ballroom dancing and water aerobics or alternatively 75 min of vigorous intensity aerobic physical activity like race-walking, jogging or running, bicycling > 15 km/h, heavy gardening (continuous digging or hoeing), swimming laps or tennis (single).

### 2.5. Diet Suggestions (both Groups “D” and “DE”)

We suggested the following diet:30–45 g of fiber per day, preferably from wholegrain products;At least 200 g of fruit per day (2–3 servings);At least 200 g of vegetables per day (2–3 servings);Fish 1–2 times per week;Saturated fatty acids to account for <10% of total energy intake, through replacement by polyunsaturated fatty acids;Trans unsaturated fatty acids: as little as possible, preferably no intake from processed food, and <1% of total energy intake from a natural origin;<5 g of salt per day;Consumption of alcoholic beverages should be limited to 2 glasses per day (20 g/d of alcohol) for men and 1 glass per day (10 g/d of alcohol) for women.

### 2.6. Statistical Analysis

Results were presented as means ± standard deviation (SD). Normal distribution was tested using the Kolmogorov–Smirnov test and statistical parametric techniques were applied.

A *t* test for paired samples was used to analyze the differences between T1 and T2 independently for each group (D, DE and total sample). Both in T1 and T2, a paired *t* test for independent samples was used to compare results between D and DE.

The in-between groups (D and DE) comparison from T1 to T2 was performed by two way mixed ANOVA (group per time).

Practical significance was assessed by calculating Cohen’s d effect size (ES). An ES (*d*) of above 0.8, between 0.8 and 0.5, between 0.5 and 0.2 and lower than 0.2 were considered as large, moderate, small and trivial, respectively. Differences between means were instead expressed as percentages. All data analysis was performed using the *MedCalc* software (Ostend, Belgium, version 12.7.0) by Schoonjans et al. [19] and significance was conventionally set at 5% (*p* < 0.05).

## 3. Results

First of all, we have just communicated that after enrolment (t0) two pts of group D and one of group DE prematurely abandoned the study for withdrawal of their consent and so they are not included in computations. Therefore, at beginning of FUP, 48 pts were in group D, while the DE group consisted of 49 subjects. Females were 47 (23 in “D” and 24 in “DE”, respectively) while males were 50 (26 in “D” and 24 in “DE”, respectively). Mean age was 62 ± 14.

Findings revealed that there was not any difference at T1 between all analyzed items (see tables’ results and ES). Thus, the starting situation was homogeneous between the groups. Analyzing T2 vs. T1 changes, we have seen that diet was able to ameliorate only WC and WHR. Indeed, associating physical activity to diet we have also obtained a significant improvement in HR, BP, BMI, lipid and glucose profiles and finally WT6M. In particular, by observing ES we have discovered a small effect on HR, triglycerides and glucose blood levels, moderate effects on SBP, BMI and HDL and finally large on DBP and WT6M. For all details, please see Table 1 and Table 2.

## 4. Discussion

We have already seen that MS is very often the first stage of obesity, diabetes, HT and so on [20]. In particular, worldwide obesity has nearly tripled in the last 40–50 years and 39% of adults aged 18 years and over (39% of men and 40% of women) were overweight in 2016, and 13% (11% of men and 15% of women) were obese [21]. Nevertheless, obesity is partly preventable.

The main cause of overweight and/or obesity is an energy imbalance between calories consumed and calories expended. Physical inactivity is due to an increasingly sedentary lifestyle [22].

Changes in physical activity and also dietary patterns are often the result of environmental and societal changes associated with development and lack of supportive policies in sectors such as health, education and food processing [23].

Therefore, healthy lifestyle promotion is crucial, so much so that it is celebrated in all the main ESC guidelines (as described in the introduction).

In our study, we have selected a common representative population of sedentary individuals; on the basis of ESC Recommendations, we have considered “sedentary” people who usually maintain an insufficient grade of exercise as fully described in guidelines and/or in aforementioned methods. All of them were diagnosed with MS. None had history of any previous CV event and so we were in the primary prevention setting. For our enrolled volunteers, lifestyle changes were appropriate and our aim was to promote exercise and diet. In particular, we have chosen a double alternative strategy, a) promoting only diet and b) both diet and exercise. In this way, we wanted to identify the most effective way for reducing CVD risk and find a simple program for fighting MS, preventing and/or delaying overt CV manifestations.

In 2019 Krankel et al. had reviewed that exercise is benefit not only for metabolic pathways but also for practical CV endpoints and they wished further studies for better delineating the topic. Our study confirmed that both exercise and diet are benefit for CV profile. In particular, by considering MS like an early CV injury if we act systematically before the onset of CV symptoms, we can really slow down CVD. Of note the first “therapeutic” approach is healthy life [24].

In our investigation, combination of diet and exercise have shown benefits in terms of blood samplings, HR, BP values and BMI in addition to WC and WHR which were improved also in group “D” (only diet).

Obviously these objectives cannot be pursued without an appropriate and valid involvement of the subject at risk. This apparently anecdotal evidence must stimulate us to promote empowerment and/or engagement of pts. An emerging concept that implies a re-organization of physicians’ work in preventing the NHS with a more constant collaboration of nurses and/or other healthcare providers [25].

In this regard, cardiac rehabilitation is increasingly at the center of discourses around CVD and secondary prevention is the new model of effective treatment of CV patients. Likewise, there is clear agreement that “rehabilitation” in primary settings is an adequate mixture of long-term programs based on diet, exercise, adherence to prescriptions (both any medications and healthy lifestyle) [26].

Our enrolled subjects, for example, were periodically interviewed by phone-calls assessing the right adherence to protocols (D and/or DE) and/or for motivating them.

On exercise, intensity and modality of training sessions were sufficiently debated. High-intensity training sessions might be unsafe, especially in untrained subjects [27]. Interval vs. continuative training choice is still a challenging topic [28]. Some studies have demonstrated similar effectiveness across all body composition measures, whilst for other authors, interval training is inefficient for the modulation of total body fat mass or total body fat percentage [29].

In the literature, we found a paper in which the authors concluded that high-intensity exercise training was more beneficial than moderate-intensity exercise training for reducing CVR in rats with MS [30].

Nevertheless, our purpose was not to define the best training program for managing MS. There was a lack of similar information in humans and thus, our choice was subordinated to the characteristics of the enlisted subjects (substantially sedentary with MS) and in accordance with exercise recommendations contained in the last ESC guidelines [17].

Moreover, in the KORA German survey, which examined people with MS, physical activity performed at regular intervals was more effective than non-regular exercise, despite intensity [31]. Those results support the hypothesis that only constant and long-lasting exercise might influence CV fitness positively, contrasting with conventional modifiable CVRF that is well-represented by MS.

Analogously, in our population, continuative exercise was able to modify some CVRF and/or positively affect performance indicators such as WT6M.

The KORA survey is also interesting for median aging of the enrolled population: not so young people and a population very similar to our sample and/or well-representative of a common scenario. Generally, the geriatric population is not in accordance with studies’ inclusion criteria but on the contrary, our pts are very often aged > 65. In our study, participants were aged from 48 to 76 years.

Nevertheless, it is well-known that the elderly are very often frail and generally frailty is an exclusion criteria [32]. Therefore, we do not have the presumption to consider our population to be totally representative of the real-life population, but at the same time, our study confirmed the benefits of both a balanced diet and constant exercise over time as also well-established by very large international registries [33], which generally might include older people.

Finally, we believe a brief comment is due also regarding blood exams’ ameliorations. In T2, indeed, both levels of glucose and triglycerides decreased, confirming data observed in other studies [34,35]. On the contrary and unfortunately, in the literature we can also find studies which affirm different evidence and thus, the topic is still quite controversial [36].

Analogously, HDL cholesterol modified with a substantial rise reflecting the beneficial effect of exercise on top of a healthy diet as already demonstrated by Bhutani et al. [37], but not by Nieman et al. [36]. Therefore, also in this case, there is not a single valid message, but a very debated reality which advises us not to provide too-hasty sentences.

## 5. Study Limitations

Our sample was large enough to obtain a “pilot” result. Nevertheless, further studies are necessary to confirm our findings. In particular, it may be more and more interesting understand and better explain a series of current unmet needs. For example, which type of exercise is more effective; besides this, is there is any substantial difference by age and/or gender and/or epigenetic changes in response to both exercise and diet or singularly to diet and exercise.

Finally, future challenges should be in also developing individualized programs in order to pursue a precision medicine and not only a prevention medicine and/or addressing which types of diet and physical activity may be more effectiveness in MS and/or in other very common pathways [38].

## 6. Conclusions

Contrasting MS with a healthy diet seems to be partially effective by reducing WC and WHR. Indeed, a comprehensive program based on a healthy diet plus regular exercise seems to be able to also modify HR, BP, BMI, blood values and WT6M. It might seem an obvious conclusion: the addition of physical activity to diet should be intuitively more profitable, but the topic is not yet full addressed in the literature and our findings will have to be confirmed by much larger population studies.

Moreover, we believe that it is pivotal that public health policies should mostly encourage a healthy lifestyle in order to achieve better results in reducing the CV burden. Moreover, all healthcare professions will have to contribute to an adherence of a “healthy life” [39,40]. In the closer future, the role of nurses and/or other allied professions such as health remote monitoring could be pivotal for maintaining regularity in healthy behaviors [41].

## Figures and Tables

**Figure 1 ijerph-17-05424-f001:**
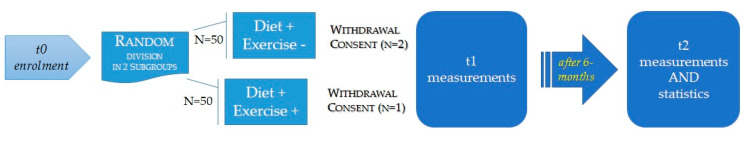
Study’s Workflow.

**Table 1 ijerph-17-05424-t001:** Main results. for total sample (*N* = 97), only diet group (D, *n* = 48) and diet plus exercise group (DE, *n* = 49).

Parameters	Group	T1	T2	d
**WT6M (meters)**	Total Sample	451.3 ± 48.8	480 ± 36.7	0.83 *
	D	453.4 ± 42.6	476.4 ± 34.4	0.81 *
	DE	450.2 ± 38.8	486.5 ± 40.1	0.86 **
	d (DE–D)	−0.19	0.79 *	-
**SBP (mmHg)**	Total Sample	133.1 ± 4.2	127.3 ± 3.8	−0.66 *
	D	132.9 ± 3.9	128.0 ± 4.3	−0.62 *
	DE	132.8 ± 4.0	126.5 ± 5.1	−0.69 *
	d (DE–D)	−0.08	−0.24 *	−
**DBP (mmHg)**	Total Sample	84.8 ± 4.0	70.9 ± 3.9	−0.82 *
	D	84.4 ± 4.0	73.1 ± 3.1	−0.81 *
	DE	85.3 ± 3.8	70.3 ± 4.2	−0.84 **
	d (DE–D)	0.13	−0.26 *	-
**HR (beats/min)**	Total Sample	75.1 ± 11.6	71.1 ± 13.0	0.27 *
	D	73.2 ± 10.7	69.4 ± 12.5	0.26 *
	DE	73.3 ± 9.8	67.4 ± 14.6	0.27 *
	d (DE–D)	−0.07	−0.19	-
**WC (cm)**	Total Sample	100.2 ± 14.7	89.6 ± 11.2	−0.45 *
	D	101.4 ± 12.6	90.4 ± 11.1	−0.44 *
	DE	100.4 ± 11.9	88.0 ± 12.3	−0.47 *
	d (DE–D)	−0.26	-0.32 *	−
**WHR**	Total Sample	0.86 ± 1.6	0.82 ± 1.9	−0.28 *
	D	0.86 ± 1.2	0.79 ± 2.1	−0.29 *
	DE	0.85 ± 1.3	0.80 ± 1.7	−0.28 *
	d (DE–D)	−0.11	0.18	-

T1 = first check; T2 = 6 months follow-up; D = only diet; DE = diet plus exercise; WT6M = 6-minwalking test; SBP = systolic blood pressure; DBP = diastolic blood pressure; HR = heart rate; WC = waist circumference; WHR = waist hip ratio. Significant differences (* *p* < 0.05, ** *p* < 0.01) between T2 and T1; d = Cohen’s d effect size. Data were expressed as mean ± standard deviation (SD).

**Table 2 ijerph-17-05424-t002:** Other results. for total sample (*N* = 97), only diet group (D, *n* = 48) and diet plus exercise group (DE, *n* = 49).

Parameters	Group	T1	T2	*d*
**FG (mg/dL)**	Total Sample	91.1 ± 11.2	83.3 ± 12.4	−0.36 *
	D	90.2 ± 10.2	88.9 ± 11.4	−0.22
	DE	92.2 ± 11.3	82.2 ± 9.5	−0.41 *
	d (DE–D)	0.07	−0.28 *	-
**HDL (mg/dL)**	Total Sample	37.2 ± 4.4	46.1 ± 4.2	0.65 *
	D	36.3 ± 5.8	38.2 ± 4.6	0.24
	DE	37.6 ± 7.0	48.3 ± 5.6	0.71 *
	d (DE–D)	0.06	0.68 *	-
**TG (mg/dL)**	Total Sample	148.1 ± 13.3	141.0 ± 12.8	−0.40 *
	D	146.0 ± 10.2	144.4 ± 11.2	−0.23
	DE	148.4 ± 9.4	139.6 ± 10.4	−0.43 *
	d (DE–D)	0.1	−0.39 *	−
**BMI (kg/m^2^)**	Total Sample	24.1 ± 4.5	21.6 ± 4.9	−0.69 *
	D	24.2 ± 4.4	23.5 ± 4.5	−0.27
	DE	24.4 ± 4.6	20.3 ± 4.8	−0.70 *
	d (DE–D)	0.05	−0.67 *	-

T1 = first check; T2 = 6 months follow-up; D = only diet; DE = diet plus exercise; FG = fasting glucose; HDL = HDL cholesterol; TG = triglycerides; BMI = body mass index; Significant differences (* *p* < 0.05, ** *p* < 0.01) between T2 and T1; d = Cohen’s d effect size. Data are expressed as mean ± standard deviation (SD).

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
