# Peer review of "Healthy Promotion for Fighting Metabolic Syndrome: Insights from Multi-Center HeRO-FiT Cohort"

_ijerph, 2020, doi:10.3390/ijerph17155424_

Round 1
Reviewer 1 Report
All together, this is a very important paper in MS.
However, the authors should fulfill the following lacunae:
The authors should redo the figure 1 because it is not in the journal quality. It is very hard to read the letters and/or numbers in their art work.
In the page 7, authors mentioned as "In literature, we had also found a paper in which authors concluded that high-intensity exercise training was more beneficial than moderate-intensity exercise training for reducing CVR in rats with the MS." However, there is no particular reference cited in the text or included in the reference list.
Conclusions should be short and crisp and so authors should rewrite the conclusion in that way.
Author Response
REVIEWER 1
All together, this is a very important paper in MS.
MANY THANKS.
However, the authors should fulfill the following lacunae:
The authors should redo the figure 1 because it is not in the journal quality. It is very hard to read the letters and/or numbers in their art work.
MODIFIED and ameliorated (FONT SIZE ENLARGED).
In the page 7, authors mentioned as "In literature, we had also found a paper in which authors concluded that high-intensity exercise training was more beneficial than moderate-intensity exercise training for reducing CVR in rats with the MS." However, there is no particular reference cited in the text or included in the reference list.
Many thanks. For a misprint I have forgotten it. Now inserted.
Conclusions should be short and crisp and so authors should rewrite the conclusion in that way.
DONE.
Reviewer 2 Report
- The introduction needs to be thoroughly revised. The authors need to expand the literature that is relevant to the study. I would suggest that the authors to present the aim of the study with regards to what is currently known, therefore highlighting the added value of the paper.
- Line 85: More information about these settings would be beneficial.
- Line 110-129: The procedure adopted for these groups should be described in as complete detail as possible.
- Figure 1 should be clearly presented.
- Line 169-179: I believe these statements should be moved to introduction.
- This statement was made without supporting reference(s) (i.e. Line 211-213).
- Limitations of the study should be clearly mentioned and discussed.
- The reference list needs formatting in accordance with IJERPH guideline.
- The standard of English needs to be improved.
Author Response
REVIEWER 2
- The introduction needs to be thoroughly revised. The authors need to expand the literature that is relevant to the study. I would suggest that the authors to present the aim of the study with regards to what is currently known, therefore highlighting the added value of the paper.
For extending a little (in our opinion, a wide enlargement could make the introduction excessively loud and heavy) the content about any current evidence, we added some references and implemented overall background (see the new text). Moreover, about your second notification, we believe that paper’s added value may be better described in Discussion/Conclusions (see current partial enlargement and/or each done modifications following all reviewers’ suggestions).
- Line 85: More information about these settings would be beneficial.
Modified.
- Line 110-129: The procedure adopted for these groups should be described in as complete detail as possible.
We believed that a more complete and/or detailed description was not possible and might be more chaotic rather than the proposed schematic version. Moreover, our protocol was not too tight for all together authors’ clear choice: we wanted to transfer a basic Prevention message (like in ESC Recommendations – see paragraph 2.4 and/or 2.5), much easier to be reached by enrolled population.
- Figure 1 should be clearly presented.
DONE.
- Line 169-179: I believe these statements should be moved to introduction.
We have inserted them in a similar position in order to start the discussion with some general considerations. But if it is pivotal for publication also by Editor, we could subsequently move.
- This statement was made without supporting reference(s) (i.e. Line 211-213).
Sure. I agree. For a misprint I have forgotten it. Now inserted.
- Limitations of the study should be clearly mentioned and discussed.
DONE with a new final paragraph after conclusions. In previous version, we decided to write only “we do not have the presumption to consider our population totally representative of real-life, but at the same time, our study confirmed the benefits of both a balanced diet and constant exercise over time as also well-established by very large international registries” in DISCUSSION.
- The reference list needs formatting in accordance with IJERPH guideline.
Yes. Many thanks. Now modified.
- The standard of English needs to be improved.
Partially DONE (see all modifications in the new submitted manuscript). Moreover, if Editor wants we may use MDPI’s specific service for that task.
Reviewer 3 Report
The manuscript “Healthy promotion for fighting …” presents the results of an investigation attempting to identify optimal treatment strategies for combatting the quickly growing disease of Metabolic Syndrome. The authors selected 100 participants (losing 3 during course of the study) to investigate the effects of one of two treatment options (diet, diet and exercise) in managing anthropometric and physiological disturbances associated with Metabolic Syndrome. For their exercise intervention, authors selected continual activity of moderate intensity, rather than interval training of greater intensity. Results indicated that both interventions had positive effects on each of the physiological variables assessed. However, when examining blood profiles, e.g. glucose, HDLs, triglycerides, the combination of exercise and diet was more effective than diet alone.
The focus and message of this work is not particularly novel but it is nice to have further scientific confirmation of the beneficial influence of exercise, especially among those stricken with Metabolic Syndrome. There are several concerns, however, that must be adequately addressed by the authors.
General Concerns
- The authors need someone who is positively fluent in the English language to edit their work.
- Tables are generally presented in submitted manuscripts as separate pages (1 table for each page), and placed after the references rather than embedded in the text.
- Please include a table presenting anthropometric information, i.e., age, gender, height, body weight, % body fat at the two time points of measurement.
- It would have been nice (and added to content of paper) to be able to compare results between men and women, as well as between aged (65 years and above) and younger adult subjects. This would provide meaningful information.
- In the Conclusions, please directly address the point of the project, that is, which intervention was most effective in managing Metabolic Syndrome.
Specific Concerns
Line Page
69-71 2 If authors are not going to discuss the details of the HeRO-FiT program here, it should not be referred to. It is like beginning a sentence without completing it. The study presented here should stand on its own.
90-95 3 More details must be presented about compliance to exercise and dietary guidelines to be followed (including caloric intake).
Tables 1, 2 It is interesting that diet alone improved all of the variables assessed in Table 1. In fact, there was no significant (statistically) difference between the two interventions. On the other hand, Table 2 shows that for blood lipids and glucose, diet plus exercise triggered significant improvements, but not diet alone. Results, sometimes surprising, from these two tables merits much more discussion and explanation.
172 6 Be careful with the statement that obesity is preventable. Many factors (physiological, psychological, genetic) contribute to obesity making its management very challenging.
180 6 How was “sedentary” defined and assessed?
204-205 6 Mention that subjects were interviewed by phone as measure of compliance in the Methods section.
208-210 6 It is mentioned that previous work has indicated that continual work and interval training are equally effective in improving body composition measures, but is this also true for other important health measures such as blood pressure, blood glucose, blood lipids, etc.?
219 7 In what ways was interval training found to be more effective than continual training?
227 7 This is a wide range of ages, you could take advantage of it by making comparisons between young vs. aged, or you should have narrowed this range when selecting participants from larger pool of candidates.
Round 2
Reviewer 2 Report
- Figure 1 still not clear enough.
- Authors should follow the journal guidelines for references (https://www.mdpi.com/journal/ijerph/instructions). For example, Author 1, A.B.; Author 2, C.D. Title of the article. Abbreviated Journal Name Year, Volume, page range.